# Comparative Study of Chitosan and Oligochitosan Coatings on Mucoadhesion of Curcumin Nanosuspensions

**DOI:** 10.3390/pharmaceutics13122154

**Published:** 2021-12-14

**Authors:** Gye Hwa Shin, Jun Tae Kim

**Affiliations:** 1Department of Food and Nutrition, Kunsan National University, Gunsan 54150, Korea; 2Department of Food and Nutrition, BioNanocomposite Research Center, Kyung Hee University, Seoul 02447, Korea

**Keywords:** curcumin, mucoadhesion, chitosan, oligochitosan, circular dichroism

## Abstract

Curcumin nanosuspensions (Cur-NSs), chitosan-coated Cur-NSs (CS-Cur-NSs), and oligochitosan-coated Cur-NSs (OCS-Cur-NSs) were prepared by using an ultrasonic homogenization technique. The mean particle size of Cur-NSs was 210.9 nm and significantly (*p* < 0.05) increased to 368.8 nm by CS coating and decreased to 172.8 nm by OCS coating. Encapsulation efficiencies of Cur-NSs, CS-Cur-NSs, and OCS-Cur-NSs were 80.6%, 91.4%, and 88.5%, respectively. The mucin adsorption of Cur-NSs was steeply increased about 3–4 times by CS and OCS coating. Morphological changes of these NSs were studied using circular dichroism spectroscopy, Fourier-transform infrared (FT-IR) spectroscopy, and transmission electron microscopy (TEM). Thus, CS-Cur-NSs and OCS-Cur-NSs showed great potential as mucoadhesive nano-carriers for the efficient delivery of water insoluble compounds like curcumin to the gastrointestinal system.

## 1. Introduction

Curcumin is one of the most widely investigated bioactive compounds due to its various functional properties, such as anti-inflammatory, antioxidant, antimicrobial, anti-cancer, and wound healing effect [1,2,3,4]. Curcumin has been also used as a traditional medicine, as well as a natural pigment in many countries [5]. Although many benefits of curcumin have been reported, curcumin has been limited in its application to food systems due to its poor water solubility and extremely low bioavailability. In order to improve its water solubility and bioavailability, many researchers have developed various micro- to nano-sized carrier systems, such as emulsions, hydrogels, liposomes, polymeric nanoparticles, solid lipid nanoparticles, and nanostructured lipid carriers [6,7,8,9].

Previous studies reported that nano-carriers could enhance the solubility and bioavailability of encapsulated core substances, and deliver components to specific regions of the gastrointestinal tract [10,11,12]. The molecular structure and composition of the nano-carrier system could be changed continuously as it passes through the gastrointestinal tract [13,14]. Especially, coatings with polysaccharides, such as alginate, chitosan, and carboxyl methyl cellulose, could influence the potential digestion of food emulsions [15,16]. Those studies also reported that polysaccharides may influence the digestion of nano-carriers because they could bind or sequester the components present in the gastrointestinal fluids, such as bile salts, mucin, phospholipids, calcium ions, fatty acids or digestive enzymes. In addition, some polysaccharides showed good adhesion properties with mucosal tissue [17].

Chitosan (CS) has been extensively studied for transporting drugs and functional foods to improve their stability and bioavailability in drug delivery system [17]. CS is a family of cationic linear polysaccharides produced by the deacetylation of chitin which is extracted from shell of crustacean. CS is composed of randomly distributed *β*(1→4)-linked *N*-acetyl-d-glucosamine and d-glucosamine units. The cationic properties of CS are useful in pharmaceutical formulations because it can make polyelectrolyte complexes with the anionic groups of most biomaterials. CS has also good biocompatibility, biodegradability, low toxicity and mucoadhesive properties [17,18]. Positively charged amine groups of CS can promote bioadhesion with negatively charged mucin. Thus, CS-coated nano-carriers can improve the electrostatic interactions with mucin in the GI tract and achieve higher absorption of functional core materials [19,20]. In addition, as a low-molecular weight chitosan, oligochitosan (OCS) is preferred for application to food systems as an alternative to natural chitosan due to its high solubility in acid-free aqueous media, as well as its biodegradability, biocompatibility, low toxicity and mucoadhesion [21].

Mucins are high molecular weighted extracellular glycoproteins produced by epithelial tissues in most animals [22,23]. Mucins have high amounts of glycosylated proteins and are highly viscous, contributing to the gel-like properties of mucus. Mucin is well known as a substrate for polymer attachment, forming the basis for their use in drug delivery [24]. Mucin is used pharmaceutically as an excipient in drug delivery, plays a vital role in preventing diseases, such as diabetes and high blood pressure, and has become an important ingredient in daily diets [24]. In contrast to chemical cross-linking, mucin can perform intermolecular and intramolecular cross-linking after spontaneous interaction with cationic biopolymers, such as CS and OCS.

The ultimate objective of this study is to improve the interactions between Cur-NSs and mucins by coating them with positively charged CS and OCS. Thus, CS- and OCS-coated curcumin-loaded nanosuspensions (Cur-NSs) were characterized and compared in terms of their physicochemical and mucoadhesive properties. In addition, the structural or conformational changes of mucin and Cur-NSs were compared after CS and OCS coating using Fourier transform infrared spectroscopy (FT-IR) and circular dichroism.

## 2. Materials and Methods

### 2.1. Materials

Curcumin (≥98%) was purchased from Fisher Scientific (Pittsburgh, PA, USA). MCT oil (mixture of 50% capric acid and 30% caprylic acid) was purchased from Now^®^ Foods (Bloomingdale, IL, USA). d-α-Tocopherol polyethylene glycol 1000 succinate (TPGS), lecithin, and mucin (extracted from porcine stomach type III) were purchased from Sigma-Aldrich (St. Louis, MO, USA). Chitosan (MW of 30,000; degree of deacetylation of 88%) and oligochitosan (MW of 3000; degree of deacetylation of 90%) were obtained from Biotech Co., Ltd. (Mokpo, Korea). Analytical grade ethanol and acetone were purchased from Duksan Pure Chemicals Co., Ltd. (Ansan, Korea). Distilled water (Milli-Q system, 18.3 M) was used for all experiments.

### 2.2. Preparation of Cur-NSs and Chitosan Coating

Cur-NSs were prepared according to the procedure described in our previous study [25]. Briefly, curcumin powder was added in a 1:2 ratio of curcumin to TPGS to an aqueous solution containing 1.25% (*w*/*v*) of TPGS and 0.5% (*w*/*v*) lecithin. The mixture was stirred at room temperature using a magnetic stirrer overnight to reach equilibrium solubility. Fully dissolved Cur-NSs were successfully fabricated by a probe type of ultrasonic homogenization with a model VCX-750 apparatus (Sonics & Materials, Newtown, CT, USA) for 30 min with 30% vibration amplitude in an ice bath to maintain a constant temperature. CS- and OCS-coated Cur-NSs were prepared by adding Cur-NSs dropwise into 0.2% CS solution prepared in 1% (*v*/*v*) acetic acid and 0.2% OCS solution prepared in distilled water, respectively.

### 2.3. Encapsulation Efficiency of Curcumin

Cur-NSs, CS-Cur-NSs, and OCS-Cur-NSs samples were centrifuged at 190× *g* for 10 min using a centrifuge (CR-21G, Hitachi High-Technologies Co., Ltd., Tokyo, Japan), and the supernatants containing Cur-NSs, CS-Cur-NSs, or OCS-Cur-NSs were separated from the pellet, which contained unloaded free curcumin. The pellet containing free curcumin was fully dissolved in methanol and used to measure the curcumin content. The quantity of free curcumin was determined using UV spectrum at 425 nm. The encapsulation efficiency (%) of curcumin was calculated using Equation (1):Encapsulation efficiency (%) = [(*C_t_* − *C_f_*)/*C_t_*] × 100(1)
where *C_t_* is the total amount of curcumin and *C_f_* is the concentration of free curcumin.

### 2.4. Mucin Adsorption Test

In order to evaluate the mucoadhesive properties of Cur-NSs, CS-Cur-NSs, and OCS-Cur-NSs, a mucin adsorption test was carried according to a previous method with a slight modification [26]. The amount of free mucin was calculated using the Bradford colorimetric method [27]. Briefly, 2 mL of mucin solution (1 mg/mL) was mixed with 2 mL of the NS samples at 37 °C for 1 h. Then, the samples were centrifuged at 10,600× *g* at 4 °C for 30 min to separate the free mucin (supernatant) from the adsorbed mucin (pellet). Diluted Bradford reagent (×5) was added to the supernatant and incubated at 37 °C in a shaking incubator at 185 rpm for 10 min. The quantity of free mucin in the supernatant was determined by measuring the absorbance at 595 nm. To make a standard curve, the absorbances of pure mucin solutions of 31.25, 62.5, 125, 250, 500, and 1000 μg/mL were also measured. Mucin adsorption (%) was calculated using Equation (2):Mucin adsorption (%) = [(*M_t_* − *M_f_*)/*M_t_*] × 100(2)
where *M_t_* is the total amount of mucin and *M_f_* is the free mucin.

### 2.5. Measurements of the Mean Particle Size, PDI, and Zeta Potential

The mean particle size, polydispersity index (PDI), and zeta potential of the Cur-NSs, CS-Cur-NSs and OCS-Cur-NSs were measured using dynamic light scattering using a nano-ZS nanosize analyzer (Malvern, Worcestershire, UK). Three milliliters of the sample dispersions were added to polystyrene latex cells, and the mean particle size, PDI, and zeta potential were measured at 25 °C with a detector angle of 90°, wavelength of 633 nm, and refractive index of 1.449. Each sample was measured at least 3 times and the average values were used.

### 2.6. Morphological Analysis

The morphological changes of the CS-Cur-NSs and OCS-Cur-NSs were investigated using transmission electron microscopy (TEM). To prepare the TEM samples, the NS samples were dropped onto carbon-coated grids and air-dried for 1 min, and the excess was removed with filter paper. This step was repeated 3 times. Finally, the grids were stained with phosphotungstic acid for 1 min and dried overnight. TEM images were obtained using a Tecnai G2 F30 TEM (Philips-FEI, Eindhoven, Holland).

### 2.7. Circular Dichroism Spectroscope Analysis

Conformational changes in mucin caused by interactions with CS or OCS were analyzed using a circular dichroism spectroscope (J-1100, JASCO International Co., Easton, MD, USA). The sample for circular dichroism was prepared by mixing the CS or OCS and mucin in a ratio of 1:1. The samples were incubated for 30 min and placed in 1 mm quartz cuvettes. Circular dichroism spectra were recorded at 25 °C from 350 to 195 nm, with a data pitch of 1 nm. A band width of 1 nm was used with a detector response time of 1 s and a scanning speed of 100 nm/min. Each spectrum was accumulated from three scans.

### 2.8. FT-IR Analysis

Structural changes of CS, and OCS after mucin adsorption were studied by Fourier transform infrared (FT-IR) spectroscopy using a model V430 apparatus (Jasco, Tokyo, Japan). The FT-IR spectra of mucin, CS, and OCS were obtained to verify the interactions between mucin and CS or OCS. Each sample was mixed with potassium bromide (KBr) at a ratio of 1:10, and prepared as a 1 mm semi-transparent pellet by compression under a force of 5 tons using a hydraulic press. Each spectrum was obtained from the average of 64 scans at a resolution of 2 cm^−1^ in the wavelength range of 4000–800 cm^−1^.

### 2.9. Statistical Analysis

The particle size and zeta potential of the NSs were statistically analyzed using analysis of variance (ANOVA). Statistical Package for the Social Science (SPSS, Version 20.0, SPSS Inc., Chicago, IL, USA) was used for this analysis. Duncan’s multiple range tests were used to determine the statistical significance among the means at 95% significant level. All results were obtained from triplicate sample analysis.

## 3. Results

### 3.1. Characterization of Cur-NSs, CS-Cur-NSs and OCS-Cur-NSs

Figure 1a shows the mean particle size and PDI of the Cur-NSs, CS-Cur-NSs, and OCS-Cur-NSs. The mean particle size of Cur-NSs was significantly (*p* < 0.05) changed after CS or OCS coatings while PDI values were maintained at less than 0.3, indicating uniform NSs. The mean particle size of Cur-NSs was 210.9 nm whereas the mean particle sizes of CS-Cur-NSs and OCS-Cur-NSs were 368.8 nm and 172.8 nm, respectively. Generally, CS coating increased particles size of micro- and nano-particles because CS molecules made thick layers on the surface of the particles. However, the small molecules of OCS might tightly interact in between lecithin and TPGS and cause the particles to shrink. This shrinking makes OCS-Cur-NS particles smaller than those of Cur-NSs. Similar results were reported in the literature [28].

A high encapsulation efficiency of bioactive compounds is necessary to exert their effects. As shown in Figure 1b, the coating with CS and OCS significantly (*p* < 0.05) increased the encapsulation efficiency from 80.6% for Cur-NSs to 91.4% for CS-Cur-NSs and 88.5% for OCS-Cur-NSs, respectively. Positively charged CS and OCS could create a strong barrier surrounding Cur-NSs particles by electrostatic interactions with negatively charged lecithin molecules. Some works in the literature have reported that chitosan coating on nano-carrier systems, such as nanostructured lipid carriers and nanoliposomes, could improve the encapsulation efficiencies in their systems [26,29].

### 3.2. Morphology of Cur-NSs, CS-Cur-NSs and OCS-Cur-NSs

Figure 2a–d shows TEM images of CS-NSs, CS-Cur-NSs and OCS-Cur-NSs. The particles of all NSs showed a round shape and uniform size without any flocculation. Particle sizes of Cur-NSs, CS-Cur-NSs, and OCS-Cur-NSs in the TEM images were consistent with those of the size analyzer results, as shown in Figure 1a. OCS-Cur-NSs showed more homogeneous distribution of the particles compared to Cur-NSs and CS-Cur-NSs.

Figure 2e shows the schematic representation of the Cur-NSs, CS-Cur-NSs, and OCS-Cur-NSs prepared in this study. Hydrophobic curcumin could be encapsulated as the core surrounded by surfactant molecules such as lecithin and TPGS. CS molecules could surround the surface of surfactant molecules in Cur-NSs and make thick layers to stabilize and protect them by electrostatic interactions between positively charged CS and negatively charged lecithin. The particle size of CS-Cur-NSs could be increased by CS coating layers. However, OCS may pull or shrink the particles by tight interactions between lecithin and TPGS. This interaction caused smaller particle sizes than that of Cur-NSs.

Figure 2f shows visual appearance of CS-Cur-NSs and OCS-Cur-NSs. Both samples look clear yellow solutions. However, CS-Cur-NSs showed a bright yellow color close to that of a lemon, whereas OCS-Cur-NSs showed an orange color. Curcumin can change color depending on changes in its structure, where keto type is formed at pH 3–7 and enol type is formed above pH 8 [30]. In this study, CS-Cur-NSs had a pH 4 because it was prepared under acidic conditions to dissolve CS, and OCS-Cur-NSs was pH 7 because OCS is easily soluble in neutral water.

### 3.3. Mucoadhesion Studies of Cur-NSs, CS-Cur-NSs and OCS-Cur-NSs

Mucus is a steric and interactive first barrier to overcome for the effective adsorption of nano-carriers to target sites. Many functional ingredients and delivery carriers showed very low bioavailability due to poor mucoadhesive properties. The development of efficient muchoadhesive delivery systems is crucial for improving the performance of functional ingredients, such as curcumin. Optimum interaction between nano-carriers and mucins is important to efficiently deliver the core materials in nano-carrier to target site and release of functional compound on gastrointestinal organs [31,32]. The mucoadhesive properties of Cur-NSs, CS-Cur-NSs, and OCS-Cur-NSs were characterized by measuring their mucin adsorption. As shown in Figure 3, the mucin adsorption by Cur-NSs was only 15.4% and significantly (*p* < 0.05) increased to 50.7% and 62.1% for CS-Cur-NSs and OCS-Cur-NSs, respectively. Although mucins are not charged at pH 2.0, they are negatively charged above pH 2.0 and positively charged below pH 2.0 due to their amphoteric properties [32]. Low mucin adsorption in Cur-NSs was expected because of the electrostatic repulsions between negatively charged Cur-NSs and negatively charged mucins in neutral pH. On the other hand, the highest mucin adsorption of 62.1% was obtained in OCS-Cur-NSs while the mucin adsorption of CS-Cur-NSs was 50.7%. Since CS solution was prepared in 1% (*v*/*v*) acetic acid, mucins in CS-Cur-NSs may not be charged or weak negatively charged in pH 2. However, mucins in OCS-Cur-NSs may have strong negative charges and make more electrostatic interactions with positively charged OCS-Cur-NSs. Thus, OCS is more efficient coating material than CS for effective curcumin delivery systems. In a previous study, it has been observed that chitosan coating on the surface of nano-carriers could be more effective in prolonging the retention time of core materials by improving the interaction with the intestinal wall mucus [26]. Mazzarino et al. (2014) also reported that chitosan-coated nanoparticles are promising carriers for hydrophobic drug delivery in the buccal mucosa because the protonated amino groups of chitosan increased the adhesion with negatively charged groups of mucin [33].

Figure 4a shows the particle size of the Cur-NSs, CS-Cur-NSs, and OCS-Cur-NSs before and after the mucin adsorption test. The particle sizes of CS-Cur-NSs and OCS-Cur-NSs were significantly (*p* < 0.05) increased from 368.8 and 172.8 nm to 428.5 and 240.4 nm, respectively, after mucin adsorption whereas the Cur-NSs was not significantly (*p* > 0.05) changed in terms of particle size by mucin adsorption. The change in particle size after mucin adsorption was also much higher for OCS-Cur-NSs (39.1%) than for CS-Cur-NSs (16.2%). This was also consistent that mucin adsorption in OCS-Cur-NSs was higher than that in CS-Cur-NSs. The zeta potential of Cur-NSs was slightly changed from −23.6 to −20.2 mV after mucin adsorption, whereas the zeta potentials of CS-Cur-NSs and OCS-Cur-NSs were greatly changed from +36.5 to −19.0 mV and +6.6 to −6.3 mV, respectively, as shown in Figure 4b. The significant (*p* < 0.05) changes in zeta potential were the result of electrostatic interactions between the carboxylate (COO^−^) or sulfonate (SO_3_^−^) groups of mucin and the amine groups (NH_3_^+^) of CS or OCS. In addition, negatively charged mucin may cover the whole surface of CS-Cur-NSs or OCS-Cur-NSs particles, as shown in Figure 4a.

### 3.4. Conformational Analysis by Circular Dichroism Spectroscope

Circular dichroism spectroscopy is sensitive to conformational changes in chiral asymmetric structures. This is useful method for the analysis of structural changes in mucin protein caused by electrostatic interactions with CS or OCS molecules. Figure 5a shows the circular dichroism spectra of pristine mucin, CS, and the complexes of CS and mucin. CS has a negative peak centered at 209 nm due to the n-π* transition of the –NHCO^−^ chromophores [34]. Mucin has an intense peak at 270 nm, which is known as the tertiary structure of mucin [35]. The complex of CS and mucin showed a broad peak from 200 nm to 250 nm indicating that CS and mucin represent their random coil conformation. The center of the circular dichroism major peak of CS was shifted from 209 nm to 220 nm and its intensity was decreased after interaction of CS and mucin. This results showed that the complex of CS and mucin had electrostatic interactions and a more coiled structure than mucin alone. The complex of CS and mucin showed a similar peak at 270 nm but lower intensity compared to that of the pristine mucin. Overall, the shift and the intense change of the complex of CS and mucin peak indicated that electrostatic interactions were formed between CS and mucin. It has been reported that the conformational change in circular dichroism spectra could be influenced by the (+/−) charge ratio of cationic and anionic materials [36].

On the other hand, OCS showed a wide peak with a weak intensity compare to that of CS, as shown in Figure 5b. OCS represented a pattern of random coil structure having a negative peak between 207 and 210 nm. The complex of OCS and mucin represented a broad negative peak with weak intensity between 197 and 230 nm, but it was not exactly halved intensity of peaks between OCS and mucin. This means that they are not just simply dispersed by two compounds but had some strong electrostatic interactions between OCS and mucin [36].

### 3.5. FT-IR Analysis of Mucin, CS and OCS

FT-IR analysis was used to verify the chemical interactions of CS or OCS and mucin. Figure 6a shows the FT-IR spectra of CS and the complex of CS and mucin. The FT-IR spectra of CS and mucin showed broad bands at 3280 cm^−1^ and 3265 cm^−1^ corresponding to the hydrogen-bonded O–H stretches and several N–H stretching bands, respectively [37]. The characteristic absorption bands of mucin were observed at 1629, and 1543 cm^−1^ assigned to C=O stretching (amide I) and N–H bending (amide II), respectively. Two peaks of CS were observed at 1636 cm^−1^ and 1580 cm^−1^ assigned to the carbonyl (C=O) stretching (amide I) and N–H bending vibration (amide II), respectively [26], as shown in Figure 6a. FT-IR analysis of the complex of CS and mucin showed a shifted C=O absorption peak at 1640 cm^−1^ (–CONH–), indicating interactions between CS and mucin. On the other hand, OCS showed very weak peaks at 1625 cm^−1^, 1592 cm^−1^ and 1524 cm^−1^ assigned to the amide I, amide II, and amide III, respectively [38]. But, the complex of OCS and mucin showed a peak at 1636 cm^−1^ assigned to the –CONH– stretching bands which also support the interactions between OCS and mucin. In addition, the absorption ratio of the amide I/amide II decreased after complex of CS or OCS and mucin as shown in Figure 6. Uthaiwat et al. [39] also concluded that strong interactions between the NH_2_ of CS and the COOH of sialic acid in mucin could reduce the amide I/amine II absorption ratio in the complex of CS gel and mucin.

## 4. Conclusions

In this study, Cur-NS, CS-Cur-NSs, and OCS-Cur-NSs were prepared using ultrasonic homogenization and coating with CS and OCS. Especially, OCS-Cur-NSs showed smaller particle size and lower PDI value than those of CS-Cur-NSs. TEM images support that OCS-Cur-NSs has smaller and homogenous particles distribution. In addition, CS- and OCS-coating improved significantly (*p* < 0.05) the mucin adsorption by electrostatic interactions between negatively charged mucin and positively charged CS or OCS molecules and the highest mucin adsorption was obtained in OCS-Cur-NSs. Our study suggested that CS-Cur-NSs and OCS-Cur-NSs might be one of the possible mucoadhesive nano-carriers to efficiently deliver insoluble compounds to the gastrointestinal system, such as curcumin. OCS has shown promise as an effective coating material for application in many food systems due to its water-soluble property.

## Figures and Tables

**Figure 1 pharmaceutics-13-02154-f001:**
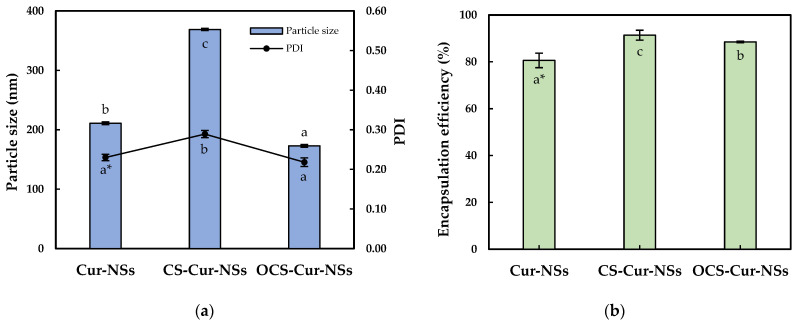
Particle size and PDI (**a**) and encapsulation efficiency (**b**) of Cur-NSs, CS-Cur-NSs and OCS-Cur-NSs. * Different letters in the same variable indicate a significant difference at *p* < 0.05. Error bars indicate the standard deviation.

**Figure 2 pharmaceutics-13-02154-f002:**
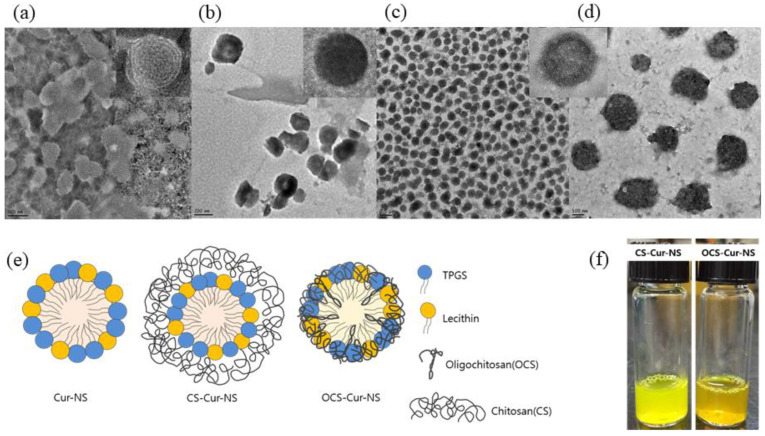
Transmission electron microscopy images of Cur-NSs (**a**), CS-Cur-NSs (**b**), OCS-Cur-NSs with scale bar of 0.5 µm (**c**) and OCS-Cur-NSs with scale bar of 100 nm (**d**). Schematic representations of Cur-NSs, CS-Cur-NSs, and OCS-Cur-NSs (**e**) and photographical images of CS-Cur-NSs and OCS-Cur-NSs (**f**).

**Figure 3 pharmaceutics-13-02154-f003:**
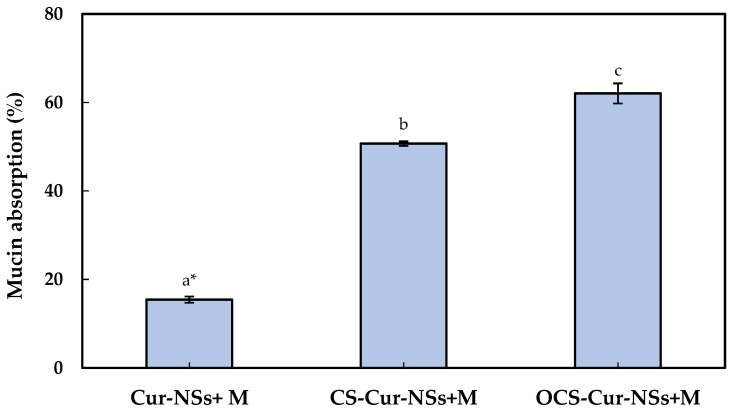
Mucin absorption of Cur-NSs, CS-Cur-NSs, and OCS-Cur-NSs after interaction with mucin protein. * Different letters indicate a significant difference at *p* < 0.05. Error bars indicate the standard deviation.

**Figure 4 pharmaceutics-13-02154-f004:**
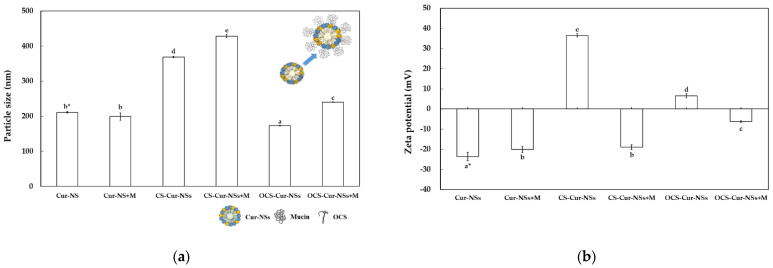
Particle size (**a**) and zeta potential (**b**) of Cur-NSs, CS-Cur-NSs, and OCS-Cur-NSs, before and after interaction with mucin protein. * Different letters in the same variable indicate a significant difference at *p* < 0.05. Error bars indicate the standard deviation.

**Figure 5 pharmaceutics-13-02154-f005:**
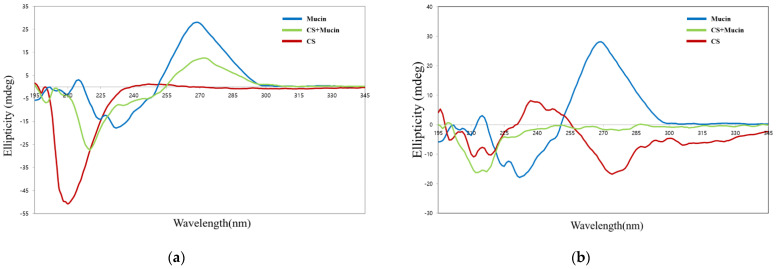
Circular dichroism spectra of CS, mucin, and CS + mucin (**a**) and OCS, mucin, and OCS + mucin (**b**).

**Figure 6 pharmaceutics-13-02154-f006:**
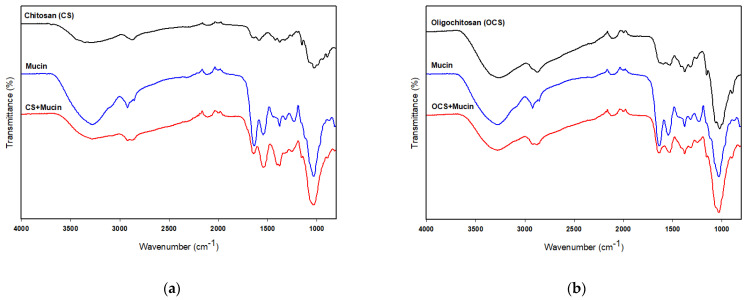
FT-IR spectra of CS, mucin, and CS + mucin (**a**) and OCS, mucin, and OCS + mucin (**b**).

## Data Availability

The data presented in this study are available upon request.

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
