# Peer review of "Comparative Study of Chitosan and Oligochitosan Coatings on Mucoadhesion of Curcumin Nanosuspensions"

_pharmaceutics, 2021, doi:10.3390/pharmaceutics13122154_

Round 1

Reviewer 1 Report

Thank you for the invitation to review this manuscript.

The authors describe a technique of coating insoluble compounds by a chitosan suspension.

The abstract is well written and contains all relevant information.

The introduction explains the background of curcumin and the reason, why it is difficult to bind. The Introduction is missing an aim or a hypothesis.

Methods are described with sufficient details; the statistical analysis is appropriate. Please describe, how descriptive analysis was done. Do you present mean and standard deviation? What is the number of samples you use for analysis (n)?

Results: Please add to your figure legends if you present mean and standard deviation or something else. For figure 1 and 3 it is not clear to me, where you found significant differences between groups. You should add this to the figure or figure legend. Could you please express the level of significance more accurate than < 0.05?

For my understanding “Results” should only present results. The comparison with the international literature should take place in the discussion. However, the article has no discussion section.

Reviewer 2 Report

The manuscript of the analyzed article has important contributions in the field of research on the bioactive compound - Curcumin- particularly attractive today in various therapeutic fields and possible solutions of its poor water solubility. The authors investigated the interactions with mucin of the chitosan and oligochitosan - curcumin particles in nanosuspensions, coated with.  The results are promising, the experimental are largely represented, but there are in the article manuscript, some suggestions and observations, from my part:

  1. Please, try to refine the Title of the article, especially in its final part.
  2. There are some style errors (ex. fourier,…)
  3. In the Introduction section, more distinctions should be made between chitosan as base and oligo-chitosan as water soluble chitosan derivative. How it influence especially the properties of the final coated curcumin particles?
  4. It is not clearly justified the use of MCT oil (mixture of 45% capric acid and 55% caprylic acid).
  5. The conclusions section must be completed with the perspectives of this preliminary studies, how could the results of this study be used for other future investigations, which is the "opening" of this research.

All the manuscript should be extensively edited for English language.  

Reviewer 3 Report

The paper by Shin and Kim is focused on the preparation, characterization, and mucoadhesive properties of curcumin nanosuspensions coated with chitosan and oligochitosan. This relatively small experimental work on the development of mucoadhesive nanoformulations of curcumin is of limited value and appears incomplete. At this point, I have to reject this paper; however, it may be improved and resubmitted after conducting additional experiments and extensive revision.

Major issues:

  1. The study of drug release kinetics is usually an essential part of the development of new drug formulations. However, no experiments on release kinetics are included in this paper.
  2. The authors speculate (lines 51-52) that "CS-coated nano-carriers can improve the electrostatic interactions with mucin in GI tract and achieve higher absorption of functional core materials”. In this regard, it is advisable to conduct experiments confirming the improved gastrointestinal penetration of curcumin in nanoformulations with chitosan and oligochitosan (for example, on the model of Caco-2 cells).
  3. The Introduction does not contain a problem statement, objective or research question, motivation, or reasons behind the study. The paper also lacks a hypothesis statement and consequently a discussion in Сonclusion as to whether the work confirmed or refuted your original hypothesis. Instead, the last paragraph of the Introduction simply summarizes the methods used in the study. This part needs to be thought through and revised.
  4. Section 3.5 FT-IR analysis of Mucin, CS and OCS is weak and very little informative. The authors focused on the description of the amide bands of mucin and chitosan, whereas amides are not charged and do not participate in the formation of a polyelectrolyte complex between these polymers and can only form hydrogen bonds. Therefore, it is premature to conclude about the interaction between mucin and chitosan based on the slight shift of the amide I band (which cannot be verified from Figure 6). A more detailed analysis of the vibrations of the charged groups of mucin and chitosan involved in the ionic interaction should be performed.

Minor issues:

  1. Throughout: Check the appropriateness of using acronyms. Overuse or inappropriate use of acronyms makes reading a paper difficult. Please note the following regarding acronyms: Acronyms should be spelled out upon first use, followed by the acronym in parentheses. Subsequently, only the acronym should be used in the text. To keep the use of acronyms to a minimum, only insert an acronym if the term is used at least three times.
  2. Throughout: Ensure the appropriate use of the number of significant figures for experimental or calculated values. In most cases, it is enough to keep 2-3 significant figures.
  3. Lines 50-53 and 56-58: This is a restatement of the same thing.
  4. Line 76: Where was the MCT oil used in the study?
  5. Line 81: Provide the degree of deacetylation for oligochitosan.
  6. Line 96: Sample/suspension compositions for determining encapsulation efficiency should be described.
  7. Figures 1,3,4: Indicate what error bars represent (standard error, standard deviation, confidence interval, etc.).
  8. The quality of Figures 3,5,6 must be improved; they are hardly visible.
  9. The English usage is often poor. Please check the paper once again to clarify meaning, improve awkward sentence structure, and correct grammar. A native English speaker with a scientific background should carefully revise the manuscript before its resubmission.

Round 2

Reviewer 1 Report

Thank you for the revision, I have no further comments.

Reviewer 2 Report

The manuscript has been sufficiently improved 

Reviewer 3 Report

The authors made some cosmetic changes to the text that did not change the overall situation. I believe that such incomplete studies should not be published in highly ranked journals like Pharmaceutics.

My major concerns are the same:

  1. The study of drug release kinetics is usually an essential part of the development of new drug formulations. However, no experiments on release kinetics are included in this paper.
  2. The authors speculate (lines 51-52) that "CS-coated nano-carriers can improve the electrostatic interactions with mucin in GI tract and achieve higher absorption of functional core materials”. In this regard, it is advisable to conduct experiments confirming the improved gastrointestinal penetration of curcumin in nanoformulations with chitosan and oligochitosan (for example, on the model of Caco-2 cells).
  3. The Introduction does not contain a problem statement, objective or research question, motivation, or reasons behind the study. The paper also lacks a hypothesis statement and consequently a discussion in Сonclusion as to whether the work confirmed or refuted your original hypothesis. Instead, the last paragraph of the Introduction simply summarizes the methods used in the study. This part needs to be thought through and revised.
  4. The English usage is often poor. Please check the paper once again to clarify meaning, improve awkward sentence structure, and correct grammar. A native English speaker with a scientific background should carefully revise the manuscript before its resubmission.